# Weibull Tone Mapping (WTM) for the Enhancement of Underwater Imagery

**DOI:** 10.3390/s23073533

**Published:** 2023-03-28

**Authors:** Chloe Amanda Game, Michael Barry Thompson, Graham David Finlayson

**Affiliations:** 1School of Computing Sciences, University of East Anglia, Norwich NR4 7TJ, UK; G.Finlayson@uea.ac.uk; 2Gardline Ltd., Prospect House, Hewett Road, Great Yarmouth NR31 0NN, UK; 3Mott Macdonald Ltd., East Wing, 69-75 Thorpe Road, Norwich NR1 1UA, UK; michael.thompson@mottmac.com

**Keywords:** underwater image enhancement, tone mapping, histogram specification, Weibull distribution

## Abstract

Domain experts prefer interactive and targeted control-point tone mapping operations (TMOs) to enhance underwater image quality and feature visibility; though this comes at the expense of time and training. In this paper, we provide end-users with a simpler and faster interactive tone-mapping approach. This is built upon Weibull Tone Mapping (WTM) theory; introduced in previous work as a preferred tool to describe and improve domain expert TMOs. We allow end-users to easily shape brightness distributions according to the Weibull distribution, using two parameter sliders which modify the distribution peak and spread. Our experiments showed that 10 domain experts found the two-slider Weibull manipulation sufficed to make a desired adjustment in >80% of images in a large dataset. For the remaining ∼20%, observers opted for a control-point TMO which can, broadly, encompass many global tone mapping algorithms. Importantly, 91% of these control-point TMOs can actually be visually well-approximated by our Weibull slider manipulation, despite users not identifying slider parameters themselves. Our work stresses the benefit of the Weibull distribution and significance of image purpose in underwater image enhancement.

## 1. Introduction

Underwater imaging is a preferred survey tool for marine environments yet optical imaging is challenging [1], particularly with respect to illumination. Within water, light attenuates with increasing depth; exponentially decreasing in intensity [2]. This is driven by increased wavelength absorption and scattering [2], resulting in low contrast images that often suffer from colour reduction and blurring effects. To lessen these effects, underwater surveys must utilise strong artificial lighting on camera platforms [3]. However, this can cause inconsistent lighting patterns. Such inconsistencies may be both deliberate, such as lighting adjustments to limit the interest of fish shoals, which can obstruct seafloor imaging and impede investigation. They can also be unintentional, like non-uniform illumination, light distortion and shadow effects. These may be caused by variation in the height or angle of imaging platforms, induced by alternative deployment methods and prevailing weather conditions. The quality of underwater images is, therefore, highly variable with low correspondence between images. Features such as seafloor-dwelling organisms may be poorly visible or irregular in appearance and, thus, annotation tasks can be severely hindered [1,4].

Tone mapping, which manipulates image histograms (or brightness distributions), can be an effective method to enhance the appearance and/or visibility of image features and suppress non-desirable lighting effects. Tone mapping can be framed as a problem of mapping an input brightness distribution to a target output distribution, such as in Figure 1. It uses a tone curve, or transform curve, which simply defines how to change image brightness values.

Tone maps such as this can be created using photo-editing tools, such as Photoshop or GIMP. In this example, a user can, effectively, make their own input to output (brightness) tone curve, see Figure 2.

At the start, a user is presented with an input to output ‘identity’ mapping; simply a graph with a line at 45${}^{\circ }$ (A). This linear tone curve means that an input brightness, of say $b_{1}$, maps to an equivalent output brightness, $b_{2}$ i.e., $b_{1}=b_{2}$. This is true for all brightness values in this case. A user can then select (B) and move (C) control-points up or down to alter the curve shape. Movement upwards causes brightness values to increase, or become brighter, whereas movement downwards cause values to reduce, becoming darker. Finally, a smooth interpolation is made between the adjusted control points to define the overall tone adjustment (D).

However, bespoke tonal adjustments such as these are time consuming. Since benthic (seafloor) surveys, for example, typically capture thousands of images there is interest in finding automatic adjustments of images. Perhaps the best known automatic tone adjustment is histogram equalization (HE). The method begins with the observation that an image that has a flat, or uniform, brightness histogram conveys maximum information (entropy) [5]. In this case, all brightness values are in use, and in an equal amount; they are equalised. This visually translates to image details appearing more conspicuous. In HE, a tone curve is found that maps an input image to an output with a flat histogram [6,7]. However, HE is too simplistic in formulation and often produces images with contrasts that are both unnaturally low and high (in different image regions). Moreover, underwater images are intrinsically dark, particularly those taken in deeper waters. The requirement for artificial lighting to capture adequate imagery leaves images dominated by a strong spot-light or halo effect. Thus, their histograms are rarely uniform and any attempts to enforce this behaviour can lead to abnormal brightness changes. In Contrast Limited Histogram Equalisation (CLHE) [8], the aim again is to flatten the histogram of the output image. However, the set of allowable tone curves are constrained so that they are neither too steep nor too shallow. CLHE has the advantage that it is a simple method and often produces improved, though not preferred, tone-renderings of images [9]. Tone mapping is, of course, a large field. In the context of underwater imaging, a range of algorithms have been applied, including [8,10,11,12,13,14,15,16,17,18]. See [19,20] for a review of underwater image processing. Since the data collected from underwater imagery, and its usage, is highly diverse, it is crucial to consider the purpose the imagery serves (and its audience) when designing, or evaluating, an enhancement. However, to our knowledge, existing automated algorithms are rarely designed from the perspective of an end-user, nor do they reliably produce outputs which are always preferred by the users.

We might say that automated algorithms are constructive, in the sense that they provide an algorithm that makes an output image without user involvement. In this paper, we consider a descriptive approach. We propose properties that an automated algorithm should have. We then adopt these properties—in a simplified user tone-manipulation scenario—without specifying how an automated algorithm should work. Rather, we manipulate the brightness values, in the same way that the data suggests an automated algorithm should alter them.

Our work builds on the prior art of Weibull Tone Mapping (WTM) [9,21]. A Weibull distribution (WD) is a smooth & unimodal function that is parameterised by two numbers that control the peak location and spread of the distribution [22,23]. In Weibull Tone Mapping, the brightness distributions of an input and a user-adjusted output image are first represented as WDs. Then, the tone map, that alters the input image to the output, is defined as the function that maps the underlying input WD to the target output WD. Significantly, for identifying benthic habitats from imagery, images tone-mapped using WTM were found to be preferred by domain experts over their own bespoke adjustments.

The prior work on WTM [9,21] did not allow users themselves to enhance images according to the WTM theory. A key contribution of this paper is that it allows such direct manipulation. First, given an input image, we can calculate its WD. Next, the user defines a target output WD by adjusting two sliders. These sliders alter the peak position and spread of the target WD. The user adjusts the sliders, whilst simultaneously viewing the live tonal alterations in the output image. They continue until a desirable enhancement is reached. In the case where users find the WTM image unsuitable, they can carry out an advanced manipulation by manipulating control-points on a tone curve.

For our study, domain-experts were tasked with enhancing images of benthic habitats such that details important for annotation purposes were more conspicuous. Broadly, we found that the two-slider WTM adjustment simplified the image enhancement task, thereby making users more productive in image annotation. Mostly, observers did not use the advanced manipulation. Yet even when they did, their output images were, almost always, found to have WD distributions. This may indicate that over-time, as users become familiar with our tool, the need to use the advanced manipulation would further diminish.

In this paper we make the following contributions:We develop the WTM algorithm into an interactive tone mapping tool to enhance underwater imagery, allowing end-users to design targeted tonal enhancements more quickly and simply.We show that arbitrary histogram adjustments using two intuitive parameters (approximating brightness and contrast) are generally sufficient to support image analysis by domain experts.We further demonstrate, on a large dataset, that the brightness histograms of underwater images can be described and adjusted very effectively using the Weibull Distribution.We demonstrate that characteristics of desirable underwater image enhancements are strongly linked to image purpose; for example, identifying and annotating specific image features.

This paper is organised as follows: in Section 2. we provide a background on tone mapping, in Section 3. we recap our WTM algorithm. In Section 4 we present a series of experiments to evaluate the performance of WTM. Lastly, in Section 5. we conclude the findings of this study.

## 2. Background

The histogram of a brightness image L(x,y), denoted h(b)=hist(L(x,y)), records the frequency of pixels of a given value *b*. It is often useful to normalise *h*, creating a probability density function (PDF), by dividing the raw frequencies of the histogram by its sum. For the remainder of this paper, we assume all histograms sum to 1 and all image brightness values lie in the interval (0,1].

Let $\underline{I}\left(x,y\right)$ denote the RGB pixel values at location (x,y) in a colour image. As defined in HSV [7], we define brightness as the maximum of R,G and B, denoted as follows: $L\left(x,y\right)=max(\underline{I}\left(x,y\right))$. Using the maximum, rather than the mean, for example, allows for simpler calculation of output colour images. Given an input colour image ${\underline{I}}_{in}\left(x,y\right)$, the corresponding maximum brightness input image is denoted as $L_{in}\left(x,y\right)$. Given an output brightness image $L_{out}\left(x,y\right)$,(created by tone mapping, for example) the output colour image, ${\underline{I}}_{out}\left(x,y\right)$ is computed as:(1)$${\underline{I}}_{out}\left(x,y\right)={\underline{I}}_{in}\left(x,y\right)\frac{L_{out}\left(x,y\right)}{L_{in}\left(x,y\right)}$$

Our brightness is the per-pixel maximum of R, G and B, ${\underline{I}}_{out}\left(x,y\right)\in \left(0,1\right]$l; that is, the maximum definition ensures that ${\underline{I}}_{out}\left(x,y\right)$ is in the display range, see [24]. In contradistinction, if we had defined brightness to be the mean of R, G and B, $L\left(x,y\right)=mean(\underline{I}\left(x,y\right))$, then the output colour image (Equation Equation 1) could have values larger than 1 and, thus, not be displayed directly. By using the maximum of R, G and B as our brightness, we avoid the question of what to do if the manipulated brightness values fall outside of the display range.

Often, in image processing pipelines (and workflows), the input brightness image is modified to make an improved output image. For example, if the details in an image are too dark to see we might brighten the image and, conversely, if the pixels are too bright we might darken the image. Both these adjustments, as well as many others, can, thus, be thought of as mapping an input image with a brightness histogram h(b) to an output image with a desired target brightness distribution $h_{targ}\left(b\right)$. The venerable Histogram Equalisation (HE) [6,7] operates in this way. Here, the target distribution, $h_{targ}\left(b\right)$, is a uniform or equalised histogram.

Importantly, there exists an increasing function of brightness t()—often called a tone map—such that the histogram of hist(f(L(x,y))) is uniform.
(2)$$L_{out}\left(x,y\right)=t(L_{in}\left(x,y\right)$$

The function, t(), is the cumulative distribution function (CDF), or integral, of the input brightness distribution; a histogram that is normalised to sum to one.
(3)$$t\left(b\right)=\int_{0}^{b}h\left(b\right)\;db$$

We show a worked example of HE in Figure 3 and Figure 4. Figure 3A,B depict an input image and its histogram-equalised counterpart, respectively. In Figure 4 we show the corresponding brightness distribution of this input image (A), labelled as ’HE’, and the HE tone curve (B)—the cumulative distribution of the input (or HE) histogram (A)—that delivers the output image. Note that a tone curve has low slopes for small and large brightness values. In the equalised image in Figure 3B, this translates to a loss of details in darker regions (i.e., bottom right) and bright regions (i.e., sponges on the cobble). In the mid-brightness range of the tone curve, the steepness of the slope causes the equalised output to appear overly enhanced, with too much contrast.

In Figure 3 we show a second enhanced image (C), possessing more detail than the input image (A), yet without the artefacts present in (B). This has been enhanced using the Contrast Limited Histogram Equalisation (CLHE) algorithm [8,10]; the tone curve of which is shown in Figure 4B. We can see from this that the tone curve never has a slope less than 0.5 or greater than 2, shown by the shaded areas. This slope-constrained tone curve is the cumulative distribution of the CLHE-derived histogram shown in Figure 4A.

CLHE strives to find a histogram $h^{\prime }\left(b\right)\approx h\left(b\right)$, such that the minimum slope (*m*) and maximum slope (*M*) of its cumulative histogram are bounded. Here, h(b) represents our input (HE) histogram and, in this case, m=0.5 & M=2, as depicted by dotted lines in Figure 4A. To understand how this is achieved, let us move to the discrete domain. We map the continuous histogram h(b) to a discrete N-vector $\underline{h}$, which has *N* bins and sums to 1. Here, the input domain (0,1] is uniformly sampled into *N* regions. As such, the *j*th bin is the interval $\left(\frac{j-1}{N},\frac{j}{N}\right]$ and $h_{j}$ is the percentage of brightness values in the image that falls in this interval; under the assumption that histograms are normalised to sum to one.

In the continuous domain, the tone curve for HE is the integral of h(), see Equation (Equation 3). In the discrete domain, we have an *N*-element tone curve, $\underline{s}$, which is analogously defined. Here the integral is replaced by a summation:(4)$$s_{j}=\sum_{i=1}^{j}h_{j}$$

As the size of each histogram bin is 1/N, the slope of the tone curve at the *j*th brightness level $s_{j}$ is defined as:(5)$$\begin{array}{cc} s_{j}=\frac{s_{j}-s_{j-1}}{1/N} & ,\;\mathrm{if}\;j>1\\ s_{1}=\frac{s_{1}}{1/N} \end{array}$$

Clearly, we can also write the slope as:(6)$$s_{j}=\frac{h_{j}}{1/N}=Nh_{j}$$

If $h_{j}=1/M$, then the slope of the corresponding tone curve (cumulative histogram) is always equal to one; that is, the tone curve is a line at 45 degrees. As we would expect, if we try and equalise an image that already has a flat histogram then the tone curve is a *null* operation i.e., each input brightness maps to the same output brightness.

In the case of CLHE, we would like to flatten an input histogram, yet limit the extent of this imposed uniformity, using a tone curve with a bounded slope that is neither too large nor too small. The CLHE algorithm achieves this by finding a proxy histogram ${\underline{h}}^{\prime }$ that is similar to the original but in which the slope conditions are adhered to. Mathematically, we would like the distance, $\left|\right|{\underline{h}}^{\prime }-\underline{h}\left|\right|$, to be small and the slope constraint $m\leq N{\underline{h}}^{\prime }\leq M$ to be met.

However, what, exactly, is meant by ‘small’ is not well defined. As discussed in [25], CLHE appears to empirically minimise a least-squares error; indeed it generally returns the same proxy as an algorithm that minimises the least-squares error. So, to a first approximation, we can consider ‘small’ to mean a minimum least-squares error. The integral of this *proxy* brightness histogram (Figure 4A) creates the CLHE tone map shown in Figure 4B.

CLHE is often deployed in its adaptive form, which is called Contrast Limited *Adaptive* Histogram Equalization (CLAHE). In CLAHE, an image is tiled into several non-overlapping regions, say a 4×4 grid. We calculate the histogram of each tile and use CLHE to derive a slope-limited tone curve. This tone curve is associated with the central pixel in each tile. Individual pixels are then mapped to output values by bilinearly interpolating the output values found for the four tone curves (from the four neighbouring centres).

Finally, the target distribution for the histogram of the output image need not be uniform in any sense. Following initial recommendations by [26], the majority of CLAHE applications to underwater images seek a target histogram modelled by a Rayleigh distribution (RD) [27].

## 3. Weibull Tone Mapping (WTM)

In CLHE, the proxy brightness distribution is related to the actual brightness distribution, where the proxy cumulative histogram (the CLHE tone curve) has a bounded slope. This implies that the proxy histogram itself adheres to the slope constraints.

Here, we wish to further constrain the shape of the proxy histogram beyond just slope-limitations. Specifically, we represent brightness histograms by proxies defined by the Weibull Distribution. To ease the exposition, we present the approach in the continuous domain. However, in practice, all computations are carried out in the discrete domain.

In its two-parameter form, a Weibull distribution is defined as:(7)$$\begin{array}{c} h_{W}\left(b;\lambda ,k\right)=\frac{k}{\lambda }{\left(\frac{b}{\lambda }\right)}^{k-1}e^{-{\left(b/\lambda \right)}^{k}},\;\;b\geq 0,\;\lambda >0,\;k>0, \end{array}$$
where *b* is the brightness value, λ is the scale parameter and *k* is the shape parameter.

Typically, RGB images are eight-bit encoded, leaving $2^{8}=256$ possible pixel values in the interval [0,255]. In this work, for mathematical simplicity, we assumed pixel values exist. ∈[0,1]. By default the Weibull distribution has a brightness scale (x-axis) that stretches to infinity. Yet, for the Weibull distributions of interest in this paper we found there are not significant Weibull probabilities for values larger than 2.55 (e.g., see [28]). We, therefore, truncated the Weibull distribution, such that b∈[0,2.55], and divided by 2.55 to scale the distribution to the [0,1] interval.

The Weibull parameters, λ and *k*, control the shape of the WD, broadly accounting for the peak position and the slope or spread of the distribution, respectively, as can be seen in Figure 5. These parameters encapsulate underwater image brightness distributions well [9,21]. It has also been demonstrated that the WD can explain the contrast statistics of natural images [29,30] and is correlated with our own perception of natural images [31].

### 3.1. Creating a Proxy Brightness Distribution Using the Weibull Distribution

Here we wish to use Weibull distributions to drive tone-mapping. The steps in our Weibull Tone Mapping approach are summarised in Figure 6. First, analogously to CLHE, we wish to approximate the histogram h(b) of a brightness image *L*, by a proxy histogram $h^{\prime }\left(b\right)$. However, here the proxy histogram is the ’closest’ Weibull distribution. The Weibull proxy, $h^{\prime }=h_{W}\left(b;\lambda ,k\right)$, is found by finding the Weibull parameters, λ and *k*, that minimise the Kullback–Leibler (KL) divergence—as a measure of closeness:(8)$$\min_{\lambda ,k}\int_{0}^{1}h\left(b\right)log\left(\frac{h(b)}{h_{W}\left(b;l,k\right)}\right)\;db$$

The KL-divergence is a probabilistic measure of the difference between the distributions h(b) and $h_{W}\left(b;l,k\right)$. If the two distributions are highly similar, then the KL-divergence is low, with 0 reached only when h(b) = $h_{W}\left(b;l,k\right)$. Thus, we seek the λ and *k* that returns the lowest KL-divergence. We approach the problem discretely, searching parameter pairs of λ∈{0,0.1,0.2,⋯,3} and k∈{0,0.1,0.2,⋯,15}. These create a large diversity of histogram shapes that fit suitably within the brightness range. In Figure 5A we see how increasing λ results in a WD in which the peak location is right-skewed, simulating a histogram of a ’brighter image. Whereas in (B), we see how decreasing *k* creates a flatter and wider histogram. If attributed to an image, this would infer higher contrast.

In [9], we have access to an input image and a user-adjusted output image, enhanced using a user-crafted tone curve (Figure 2). Thus, for input and output histograms, $h_{in}\left(b\right)$ and $h_{out}\left(b\right)$, we find the Weibull proxies, $h_{in}^{\prime }\left(b\right)$ and $h_{out}^{\prime }\left(b\right)$, by minimising KL-divergence. This results in proxy WDs that closely *match* the input and output brightness distributions, as demonstrated in Figure 7A, in solid and dashed lines, respectively.

### 3.2. Calculating the Weibull Tone Map

The tone curves that map the input and output Weibull proxy distributions to a uniform brightness histogram are $t_{in}\left(b\right)$ and $t_{out}\left(b\right)$, respectively, as seen in Equation (Equation 3). Where the corresponding CDF (or cumulative histogram), of a WD $h_{W}\left(b\right)$, is the tone mapping function t() (as with HE). It is denoted as:(9)$$t_{W}\left(b;\lambda ,k\right)=1-e^{-{\left(b/\lambda \right)}^{k}},\;\;b\;\in \left(0,\infty \right),$$

The inverse of these tone curves, denoted as $t_{in}^{-1}\left(b\right)$ and $t_{out}^{-1}\left(b\right)$, therefore, map the uniform distribution to the input and output proxies. In WTM, we map the original input brightness image $L_{in}\left(x,y\right)$ using the tone curve t(), so that the input proxy brightness histogram matches the output proxy. We apply:(10)$$t\left(b\right)=t_{out}^{-1}\left(t_{in}\left(b\right)\right)$$

This results in a tone map that closely approximates that created by an analyst, as shown in Figure 7B.

In Figure 8, the corresponding images are shown, demonstrating that application of the WTM tone map to image (A), results in image (C), that is almost indistinguishable from the user-adjusted output image in (B). The tonal adjustments in this case produced a slightly darker, but better contrasted, output image. Visibility of textural details improved, aiding assessment of substrate complexity and, thus, annotation of the habitat type. In general, over a large set of user-adjusted images, approximating tone curves in this way generated outputs that were visually similar. Moreover, to the extent that there are visual differences, these were ‘liked’ by users. Indeed, Weibull Tone Mapping delivered outputs that were slightly preferred by observers, compared to the images generated by the users themselves [9].

### 3.3. WTM as a Parameterised Enhancement Tool

In our WTM method, we approximate input and output brightness histograms, ($h_{in}\left(b\right)$ and $h_{out}\left(b\right)$), by proxies that follow the Weibull distribution ($h_{in}^{\prime }\left(b\right)$ and $h_{out}^{\prime }\left(b\right)$). Then, the tone mapping, t(), that is applied to the input brightness image is the curve that maps the input to output proxy. The output brightness image is mapped from the input, according to ${\hat{L}}_{out}\left(x,y\right)=t\left(L_{in}\left(x,y\right)\right)$.

In the context of the parameterised tool we develop here, we continue to use the Weibull proxy for the input brightness distribution. However, the behaviour of the output distribution is determined by the user, by adjusting the two parameters that drive the Weibull distribution. In effect, they define the output proxy distribution. An analyst can simultaneously look at the tone-mapped output image while continuing to adjust the Weibull parameters until a preferred result is found.

Under the hood, the users adjust the Weibull parameters, λ and *k*, as seen in Equation (Equation 7). A high λ is associated with a histogram peak towards the brightest region in the dynamic range of the image, whereas a high *k* decreases the slope, narrowing the peak and its spread across the same dynamic range. Thus, increasing λ in ${\hat{h}}_{out}\left(b\right)$ results in a WTM output image, ${\hat{L}}_{out}\left(x,y\right)$, that is brighter than its input, $L_{in}\left(x,y\right)$. Here, we used the $\hat{.}$ notation to indicate a user-defined proxy). If we select a lower *k* than the input WD for the target WD, this causes ${\hat{L}}_{out}\left(x,y\right)$ to appear to have more contrast than $L_{in}\left(x,y\right)$, with increased visibility of edge pixels and textures. In Figure 8D–F we show a range of possible output images following the parameterised WTM.

## 4. Experiments

In [9,21], we established that WTM delivered preferred tonal enhancements for analysts, in the sense that the enhanced images helped them to identify marine benthic habitats from imagery. Here, we explore the usability and performance of WTM implemented as a parameterised tone mapping tool. We investigate whether users find this approach sufficient, compared to classic and manual control-point tone-mapping approaches, that are more complex. The experimental design and the results are summarised in the following sections.

### 4.1. Domain-Expert Tone-Mapping

We asked 10 image analysts at our collaborator Gardline Ltd. (Great Yarmouth, UK) to tonally adjust underwater images, using our bespoke GUI (Figure 9), so that details required to annotate the content of the image (i.e., the habitat) were made as conspicuous as possible. Participants were instructed to find a suitable WTM enhancement by manipulating two sliders that modified the parameters, λ & *k*, of the target output WD. For user clarity, λ & *k* were respectively named as *Brightness* and *Contrast* in the GUI. These terms have two advantages. First, they are intuitively understood by the users; appearing in almost all image adjustment tools. Second, the effect of WTM on an image often resembles a sort of brightness and contrast change. This is entirely to be expected as the WD parameters, λ and *k*, broadly map to the terms brightness (peak of a distribution) and contrast (distribution width or spread). Additionally, as explained in WTM (Section 3), the WD of an input image is mapped to the WD of a target and these distributions are described in terms of their peak/brightness and width/contrast. In effect, the tone map that modifies one WD to another is effectively making a brightness and contrast adjustment (by definition).

When a suitable WTM (contrast and brightness) adjustment could not be found, an analyst could construct custom tone mapping, by pressing *Advanced*. This action would plot the current WTM tone map that they could then modify using 6 control points fixed along the x-axis at c∈[0,0.2,⋯,1]. Only movements that maintained a monotonically increasing tone map were possible. Movements to each control point created a new tone map using a Piecewise Cubic Hermite Interpolating Polynomial [32], or PCHIP interpolation, through each point. All tone maps in this study were vectors of 256 values.

Each analyst received GUI training on how to make WTM and custom tone mapping adjustments, with separate test data, before conducting the experiment. Following this training, they were each asked to adjust (enhance) a unique dataset of 42 RGB JPEG images (3236 × 4320), randomly selected from a larger underwater image dataset provided by Gardline Ltd. The random selection contained images of 6 broad habitat classes (7 images per habitat), representing the breadth of biological and physical features expected in the Gardline dataset. These classes are summarised in Table 1 and image examples of each are shown in Figure 10. As we had 10 analysts, their adjustments resulted in 42 × 10 = 420 unique image enhancements. Augmenting this set, a further *common* sample set of 18 images (3 per habitat class) was shared with each analyst. Each analyst viewed the common sample set twice, creating a further (18 × 2) × 10 = 360 enhancements. This allowed us to investigate the intra- and inter-person variability of the tonal adjustments.

This study was conducted under ISO standard 3664:2009 conditions [33]; with participants sitting approximately 70 cm from the display in a darkened room. On average, analysts took ∼30 min to complete the experiment.

### 4.2. WTM Suitability for Underwater Imagery

Domain experts in this study overwhelmingly used the 2-slider WTM to adjust the underwater imagery, with 81% of the total image dataset satisfactorily enhanced without recourse to the control-point tone curve adjustment. Moreover, on average, individual observers selected WTM for 81% (±19) of their images. In fact, 60% of observers chose WTM almost (≥90%) exclusively in their images. This would suggest that tonal adjustments offered by WTM are highly suitable for enhancing underwater imagery, in accordance with [9,21].

Observers in this study designed their bespoke enhancements to improve image quality to aid image annotation. These enhancements were subjective and tailored and, thus, might not adhere to aesthetic improvements in the conventional sense, such as those that score highly with objective reference metrics. Indeed, studies have shown that objective assessments of image quality do not always correspond well with subjective perception [34,35]. That being said, we found the enhancements in this study were also beneficial objectively. In Table 2 we present the following three popular no-reference quality metrics for underwater image enhancement: (1) Underwater Image Quality Metric (UIQM) [36], (2) Underwater Color Image Quality Evaluation (UCIQE) index [37] and (3) a colourfulness, contrast and fog density (CCF) metric [38]. The UIQM evaluates image colourfulness, sharpness and contrast, whereas UCIQE measures the degradation of colour in the CIELAB colour space. CCF is inspired by underwater imaging absorption and scattering characteristics to predict colour loss, blurring and fog/haze. For each, better image quality is associated with a higher value. Image quality was, thus, improved in enhanced output images, though not significantly in terms of UCIQE (colourfulness). However, this is unsurprising, given that the higher values of this metric are associated with a more colourful image, yet the tonal adjustments in this study preserved chromaticity. Assessing the output metrics in more detail, we see that the limited cases of custom control-point adjustments improved image quality more, on average, than the WTM-enhancements. However, in the following experiment we show that this was independent of the tone mapping method and linked to the enhancements themselves.

Intra-observer variability in this study was low for image pairs in the common set, see Table 3. In row *Agreement*, we detail the proportion of times analysts used the same tool (WTM or control-point) for each of the 18 images in the control set, which they viewed twice. This showed that they typically selected the same enhancement tool, with 80% of analysts choosing the same tool in >90% of their common images. Mean intra-observer agreement was, therefore, also high, at 93% (±8).

In the remaining rows (Table 3), we consider the average *relative* extent to which the observers’ tool parameters varied between image pairs when they selected the same tool. Parameters here refer to λ & *k* for WTM-enhanced images or each of the control-points in the interactively tone-mapped images. We compared variance between observers and parameters using a coefficient of variation (CV), or normalized standard deviation. CV is a standardized measure of dispersion around the mean, calculated as $CV=\frac{\sigma }{\mu }$, where σ is the standard deviation and μ the mean. A CV of zero indicates zero dispersion from the mean and, thus, equal values.

For each observer, we determined the CV for each parameter (in each image pair) and summarised by the mean across image pairs (MCV). In WTM-enhanced pairs, we denoted this as $MCV_{\lambda }$ and $MCV_{k}$ in Table 3. For simplicity and brevity, we do not report the MCV for each control-point (or tone-map parameter). Instead, we first derived the average control-point CV in each image pair (creating one summary measure) and, then, as before, derived the mean of this across image pairs, denoted as $MCV_{c}$ in Table 3. Lastly, if an enhancement tool was not used by an observer, we denoted the absent MCV values by ‘-’ in Table 3.

When using WTM, analysts typically introduced more variation between image pairs, in terms of *k*, a proxy for *contrast*, than λ, a proxy for *brightness*. Comparatively, there was more variation in control-point adjustments between images, with the exception of observer 10 whose control-points varied by MCV = 0.06, on average. Increased MCV is unsurprising with the control-point tool, as the potential for difference increases with more parameters (or control-points); here requiring 6 compared to 2 in WTM. These results demonstrated that WTM parameters were more similar and, thus, analyst adjustments were more consistent.

Variability between observers (inter-personal) is particularly likely in studies such as this, as observers manipulate images according to their own aesthetic preferences. In terms of tool preference, we found, on average, that 81% (± 4) selected the same enhancement type across images; specifically, the favoured tool in the dataset, WTM, see Table 4. There was a comparable proportion to the full dataset.

Using MCV, we assessed similarity of selected parameters across images for each tool. Note that here MCV refers to the average across all CV values i.e., across all observer image pairs. As with intra-personal variance, we found that observer adjustments using WTM were more similar across images than those enhanced with the control-point tool. For images enhanced with WTM, observers were again more varied in their selection of *k*, than λ. For the remaining images, control-point adjustments were, comparatively, more variable on average. Assessing the MCV in all of these cases provides only relative comparison. It is not possible to say what is a *good* value. However, a value closer to zero suggests higher similarity of observer adjustments when using WTM. This general behaviour of end-user WTM preferences is promising for future development of an automatic WTM.

### 4.3. Simplification of Control-Point Tonal Manipulations

In previous work [9], it was found that, often, an analyst-defined control-point tone curve could be well approximated by WTM. That is, the control-point tone curve could be interpreted as the tone adjustment that maps a Weibull approximation of the input brightness distribution to the Weibull approximation of the control-point-adjusted brightness distribution. Thus, we wondered if the same result would be found here in the case of control-point adjusted images. In this paper, when an analyst could not obtain a suitable output using the WTM slider adjustment, they resorted to a custom control-point manipulation. Here, we ask whether—like the prior experiments—these custom adjustments might also be interpreted as a WTM. If this was found to be the case this would imply that the WTM sufficed in general, but that finding the best WTM could not always be easily found using the two slider adjustment.

In Table 5, we summarise the colour difference between WTM approximations and their custom-enhanced counterparts. We adopted the CIELAB Delta E (or $\Delta E_{ab}^{*}$) as a measure of colour difference. First, we converted a custom-adjusted RGB image to its corresponding CIELAB (CIE 1976 $L^{*}a^{*}b^{*}$) image, where, at each pixel, we had a *L*, *a* and *b* triplet. Describing CIELAB is beyond the scope of this paper, but suffice to say that *L* represents brightness, whilst *a* and *b* encode the chromatic aspects of an image, see [39]. Second, we calculated the CIELAB image for the WTM approximation. Denoting dependence on the custom- and WTM-adjusted images using the subscripts ${}_{c}$ and ${}_{w}$, respectively, the $\Delta E_{ab}^{*}$ difference for one pair of corresponding pixels is calculated as:(11)$$\Delta E_{ab}^{*}=\sqrt{{\left(L_{c}^{*}-L_{w}^{*}\right)}^{2}+{\left(a_{c}^{*}-a_{w}^{*}\right)}^{2}+{\left(b_{c}^{*}-b_{w}^{*}\right)}^{2}}$$

Significantly, CIELAB was designed to be a perceptually uniform space that correlates with perceived colour difference [39]. In that regard, a $\Delta E_{ab}^{*}$ of approximately 1 coincides with a *just noticeable difference*. In terms of images, if the average $\Delta E_{ab}^{*}$ calculated between images (across all pixel pairs) is up to 5, then images appear the same, or similar [40].

We found that the average colour difference between our WTM- and custom-adjusted image pairs was small, with an average $\Delta E_{ab}^{*}$ of 2.98 (±0.52). Furthermore, a significant proportion (91%) had a mean $\Delta E_{ab}^{*}$< 5, demonstrating that, for the most part, custom tone maps could be convincingly approximated using WTM. In Figure 11 we show example images that were tone-mapped by an expert, and their WTM approximations. The colour differences between each pair was <5, and, in each case, the differences were near indistinguishable to the observer. These results indicated that when a control-point adjustment was made (∼20% of adjustments) the adjusted images could be well-approximated by the WTM model (i.e., the closest WTM adjustment resulted in a similar image visually). Additionally, these successful WTM approximations (mean $\Delta E_{ab}^{*}$< 5) did not jeopardize image quality in an objective sense, with metrics equivalent to their custom counterpart, see Table 6.

Custom tone maps that could not be well matched by WTM were varied in behaviour, see Figure 12. In general, they appeared to be complex operations, with, sometimes, multiple transitions between very low and steeper gradients. This is a morphology that is not consistent with WTM tone maps, which are smoother and simpler. Aesthetically speaking these tone curves can, counter-intuitively, lead to too little and too much contrast in parts of an image and cause it to appear unnatural. However these *outlier* tone adjustments **were** preferred by the analysts as they helped them to see details important to identify the image content; in this case, the habitat. Thus, they did not need to be aesthetically pleasing. That said, there were few outliers and only 9% of control-point tone maps were not well-approximated by the Weibull approximation method. Combining the successful WTM approximations (mean $\Delta E_{ab}^{*}$< 5) with the observer-selected WTM enhancements, we found that, for >98% of image adjustments, there was a suitable WTM to enhance the image in a way that was useful for the analyst. This offers the promise of supporting analysts to make quicker adjustments of their images, with control-point tone adjustments rarely required.

### 4.4. Behaviour of WTM Enhancements

The experiments demonstrated that the majority of analysts preferred Weibull Tone Mapping to support their benthic image analyses. In this section, we discuss these enhancements in more detail, contextualising them by considering the influence of image content. Note that, when describing these tonal adjustments, we also include the WTM approximations that had a mean $\Delta E_{ab}^{*}$< 5.

In general, analysts decreased λ between input and output Weibull Distributions in the experiment (61% of images). Decreasing λ (a proxy for brightness) shifts the peak of a brightness histogram towards a lower pixel intensity. This adjustment, thus, causes a brightness reduction in images following WTM. This may have helped to lessen the intensity of the light cone and halo-effect, which are common problems in artificially lit underwater imagery.

Of the images that were darkened, 45% were classified as soft substrate (SS) habitats (SS1 & SS2), as described in Table 1. In Figure 13A we show the mean standardized (Z-Score) difference in WTM parameters, between input and output WDs, for each habitat class in the image dataset. Note that the differences presented for average λ refer to cases where only λ was decreased. The full results are presented in Appendix A. From this, we note that not only did SS habitats 1 & 2 represent a large proportion of darkened images, but the extent to which they were darkened was larger than average. In Figure 13B we detail the average standardised (Z-Score) λ value across all input WDs for each habitat class. From this, we see that SS images in this dataset were significantly brighter, on average, demonstrated by a high λ value, likely due to the light and homogeneous appearance of sand and mud. Decreasing λ, in output images, may have helped reduce the bright illumination (reflective) effects on the seafloor, whilst highlighting the appearance of burrows (a distinguishing habitat feature).

Of the remaining images (55%), in which λ was decreased, the other habitats (HS1, HS2, Cor1 & Cor2) were represented roughly equivalently; on average, ∼14% each. Yet the extent to which λ was reduced in each was variable, see Figure 13A. Most notably, the brightness of images classified as hard substrate (HS) (HS1 & HS2, Table 1), on average, reduced significantly less. Images of these more structurally complex habitats were significantly darker than SS images (Figure 13B), containing increased shadow presence around topographic features, such as boulders and cobbles. Small reductions in λ would help to diminish the unwanted brightness effects mentioned prior, yet minimise visibility reduction of distinguishing features within shadowed/darker image regions. Reef framework (Cor1) and soft coral (Cor2) habitats (Table 1) can also feature such shadowing, due to their pronounced height above the seafloor, in association with topographic highs. Yet these images were, on average, darkened to a greater extent than HS habitats. Corals, such as stony coral *Desmophyllum pertusum* in habitat Cor1 and soft coral *Paragorgia arborea* in habitat Cor2, typically appeared very bright (Figure 13B) and sometimes dominated the field-of-view. A logical enhancement may, therefore, be to reduce brightness. This is particularly true when increasing contrast, as the intensity of pixels in the regions of interest are elevated and potentially saturated.

Intensifying contrast is an important tonal adjustment in underwater images [11,12,41], to lessen undesirable image effects induced by light absorption in the water medium and scattering due to particulates (i.e., *marine snow*) [19]. It is no surprise, therefore, that contrast was enhanced in 74% of images. Analysts achieved this with WTM by decreasing *k* (a proxy for contrast), which flattened the peak of the brightness histogram. In 43% of images in which *k* was decreased SS habitats (SS1 & SS2) were contained. This reduction in *k* was also significantly greater for these habitats on average, as shown in Figure 13A. Images of SS habitats were the most poorly contrasted in this work, indicated by the high *k* values in Figure 13B. Images typically lacked edge details and contained more cryptic fauna, such as those that burrow into the sediment. Increasing contrast would, therefore, allow analysts to improve seafloor texture visualisation; in this case, obtaining absence of gravels and pebbles in order to classify an SS habitat. Distinguishing between habitats SS1 & SS2It would also allow easier searching for sponge presence; a factor equally applicable to discriminating between HS habitats (HS1 & HS2), which represented 37% of *k* decreased images. However, as evident from Figure 13A, contrast was increased significantly less than average in HS images. HS images, as well as the soft coral and reef images (Cor1 & Cor2), typically appeared better contrasted (Figure 13B) and highly textured. The degree to which *k* was reduced in each of these was, therefore, likely to be less. Few soft coral and reef images received a contrast enhancement by analysts and those that did were adjusted significantly less than average. Enhancing contrast in these images, thus, appears less important to analysts. In fact, of the 26% of images in which a contrast reduction was enforced, by increasing *k*, 66% were classified as soft coral or reef.

A final remark on the behaviour of the WTM adjustments was that 17% compressed the dynamic range. Shrinking the dynamic range reduces contrast, but can, however, also be used to lessen the intensity of stark bright and dark regions. It is no surprise, therefore, that this dynamic compression mostly occurred in images containing soft corals (51%) and Reef (38%). These images are typically dominated by bright corals in the image foreground, surrounded by a very dark background. Reducing these intensities would allow for better visualisation of cryptic features.

As expected, these results demonstrated that the types of tonal adjustments made by end-users are logical. They are clearly driven by the underlying brightness distributions of underwater images, which, in turn, are a product of artificial lighting interactions with the seafloor. For example, a positive trend exists between the brightness of images and the extent to which they are darkened, as well as a negative trend between general seabed complexity and the degree of contrast enhancement.

## 5. Conclusions

Building on our prior work, we showed here that the Weibull distribution is highly suitable for both modelling, and adjusting, the brightness histograms of underwater images. Its properties are driven by two reasonably intuitive parameters which roughly conform to the brightness (peak) and contrast (spread) of a distribution. These preserve the natural behaviour of pixel intensities in underwater imagery well, but can also provide enhancements to support annotation through their modification using our Weibull Tone Mapping (WTM) algorithm.

We demonstrated, here, that the characteristics of user tonal enhancements are tightly linked to the content of underwater imagery and their associated brightness distributions, and the purpose the imagery serves (in this case habitat annotation). This explains the increased preference amongst analysts for bespoke tone-mapping adjustments over more general automatic tonal enhancements identified in prior work. Although time-consuming, control-point tonal manipulations offer end-users more complex and targeted tonal enhancements. That being said, this work showed that, given the choice, annotators rarely opted for control-point tonal manipulations. Instead, they preferred to utilise WTM, specifying desirable Weibull brightness distributions by simply manipulating its two parameter-sliders. Furthermore, in the few cases where a control-point tone map was sought, the majority behaved like WTM.

Since most analysts enhance imagery according to the Weibull distribution, WTM is a useful mechanism to grant analysts the ability to modify an image, such that it maintains Weibull properties. It also strikes a good trade-off between the flexibility of bespoke control-point manipulation and the simplicity and speed of an automated tone-mapper. Thus, it can easily be used as a *live* modification tool alongside annotation. WTM also lends itself well to future automation, by highlighting properties that an automated enhancement should have to support underwater image analysis. We note that. although the design and current function of WTM is domain-specific (underwater imaging), its usage could extend outside of this scope. For example, future work could investigate its performance in medical imaging or images collected for terrestrial- or aerial-based ecological surveys.

## Figures and Tables

**Figure 1 sensors-23-03533-f001:**
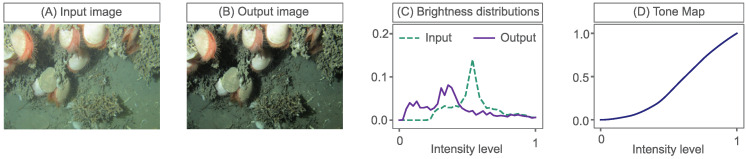
An example of tone mapping: (**A**) an input image, (**B**) an enhanced output image, (**C**) their corresponding brightness distributions and (**D**) the tone map that matches the input histogram to the output histogram.

**Figure 2 sensors-23-03533-f002:**
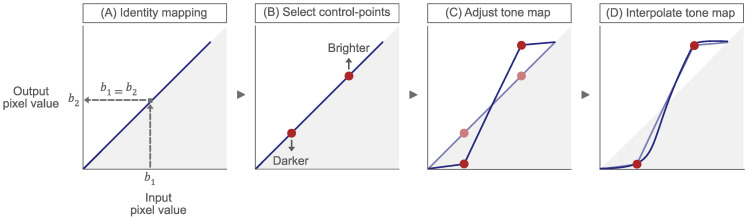
A diagram of tone map creation and interactive manipulation: (**A**) an input to output ‘identity’ mapping, (**B**) selection of control-points to modify the mapping (**C**,**D**) a smooth interpolation to create the final tone map.

**Figure 3 sensors-23-03533-f003:**
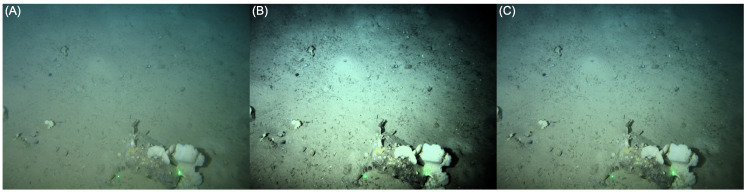
Image enhancement example: (**A**) Unenhanced (original), (**B**) HE and (**C**) CLHE.

**Figure 4 sensors-23-03533-f004:**
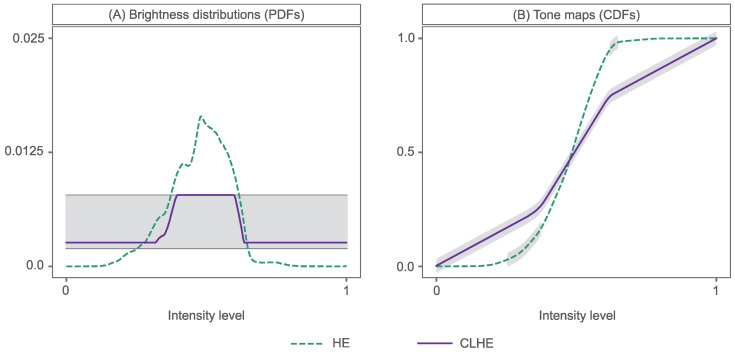
HE & CLHE tone mapping: (**A**) An Input PDF (HE proxy histogram) and its slope-limited (CLHE) proxy, (**B**) their respective CDF’s or tone maps. Grey lines in (**A**) depict upper and lower slope bounds of 2/L and 0.5/L respectively, where L=256 bins. Grey shading in (**B**) highlights area of each tone map that fall within the slope thresholds.

**Figure 5 sensors-23-03533-f005:**
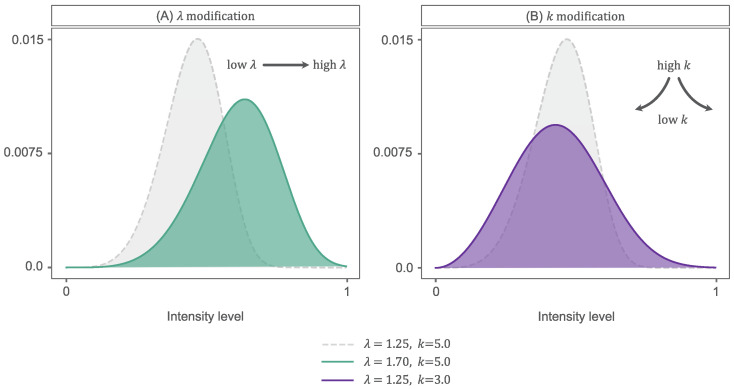
Parameter control over WD properties: (**A**) λ modification and (**B**) *k* modification.

**Figure 6 sensors-23-03533-f006:**
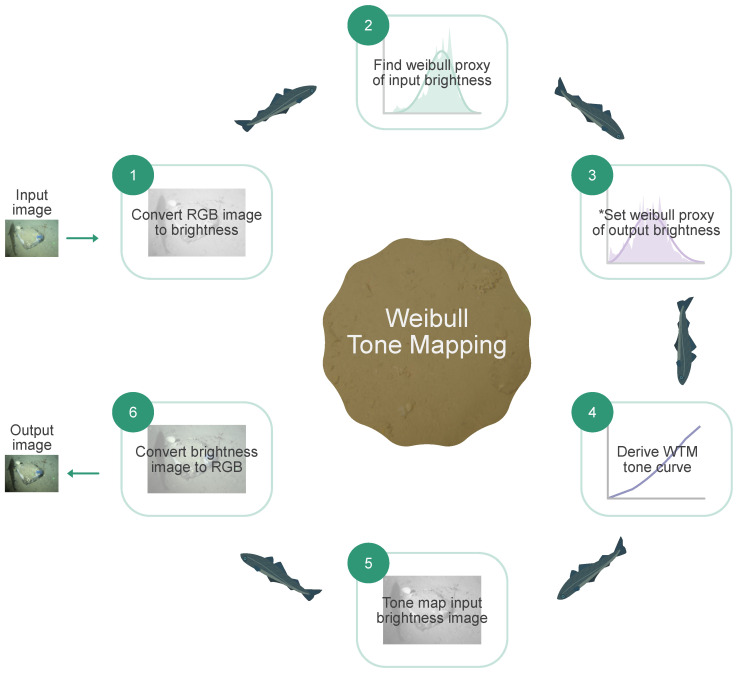
Weibull Tone Mapping flow chart. * Note that, in Step 3. we specify parameters (λ & *k*) to create a desired WD (Equation (Equation 7)). This creates an original WTM enhancement. However, Step 3 can, alternatively, approximate an existing tonal enhancement from its adjusted brightness distribution (Equation (Equation 8)). Further description of the flow chart steps is provided in text.

**Figure 7 sensors-23-03533-f007:**
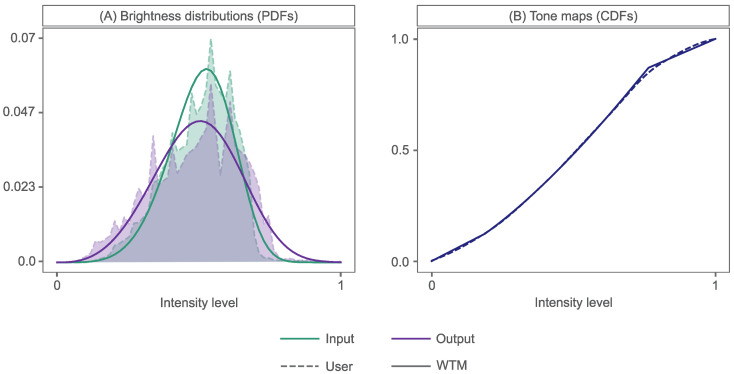
WTM method showing (**A**) WTM proxies of an Input and Output (user-adjusted) brightness PDF and (**B**) the corresponding tone maps that transform PDFs from Input to Output. Dashed and smooth lines depict User and WTM respectively.

**Figure 8 sensors-23-03533-f008:**
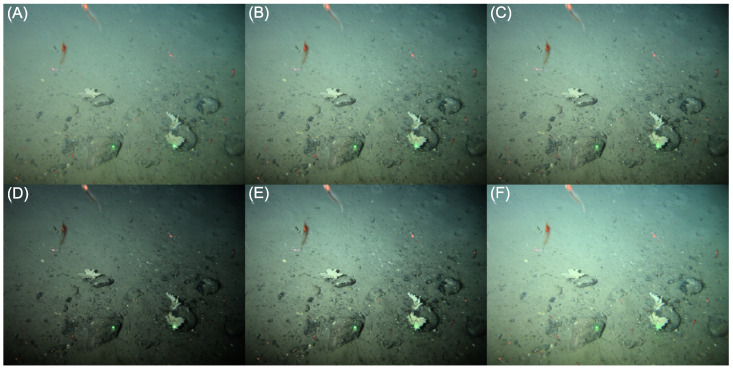
WTM example: (**A**) Input image, (**B**) User-adjusted Output, (**C**) WTM approximation of Output image (**B**) and (**D**–**F**) parameterised WTM adjustments of (**A**).

**Figure 9 sensors-23-03533-f009:**
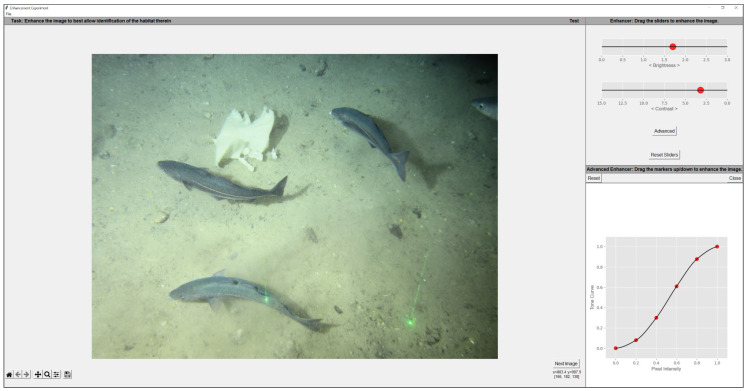
Experiment GUI.

**Figure 10 sensors-23-03533-f010:**
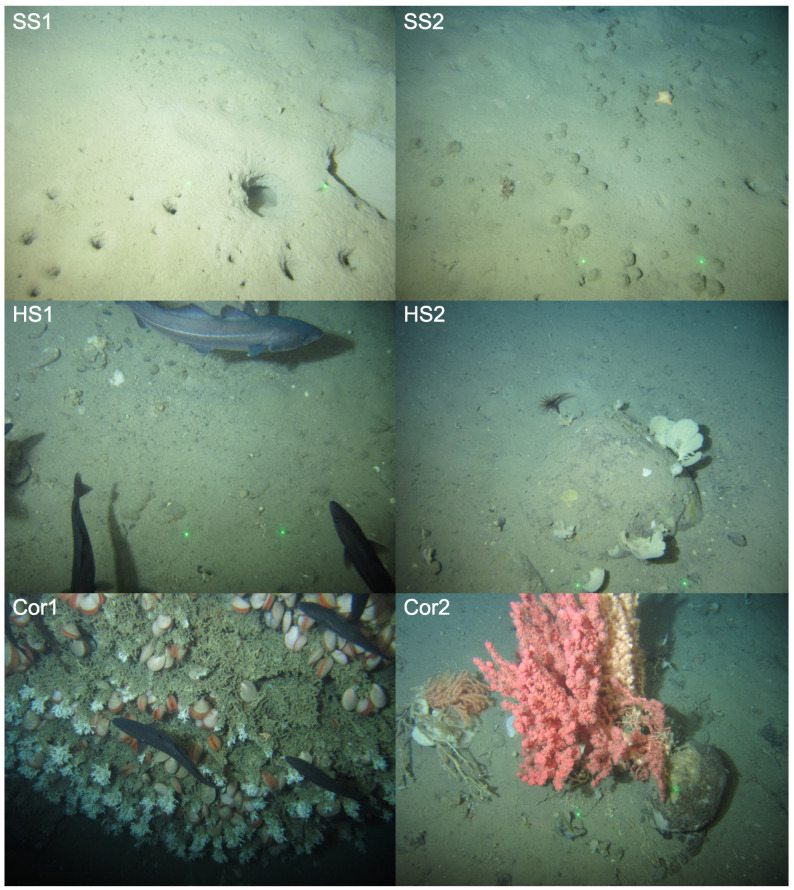
Example of Habitat classes in image dataset: SS1 (Soft Substrate), SS2 (Soft Substrate Sponge Community), HS1 (Hard Substrate), HS2 (Hard Substrate Sponge Community), Cor1 (Reef Framework) and Cor2 (Soft Corals). See Table 1 for further class details.

**Figure 11 sensors-23-03533-f011:**
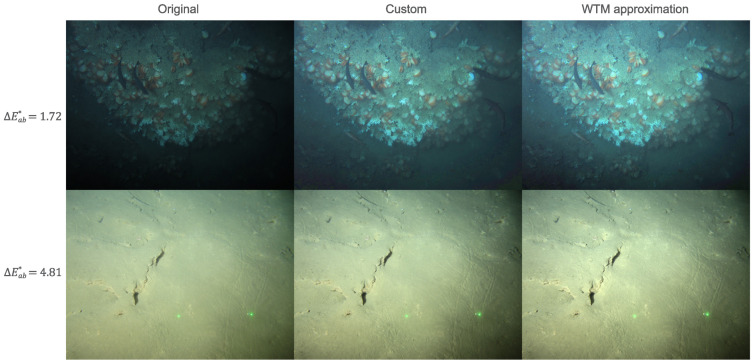
Example of WTM approximation of custom control-point adjustments. Colour difference between the custom and WTM image is summarised by mean $\Delta E_{ab}^{*}$ across pixel pairs.

**Figure 12 sensors-23-03533-f012:**
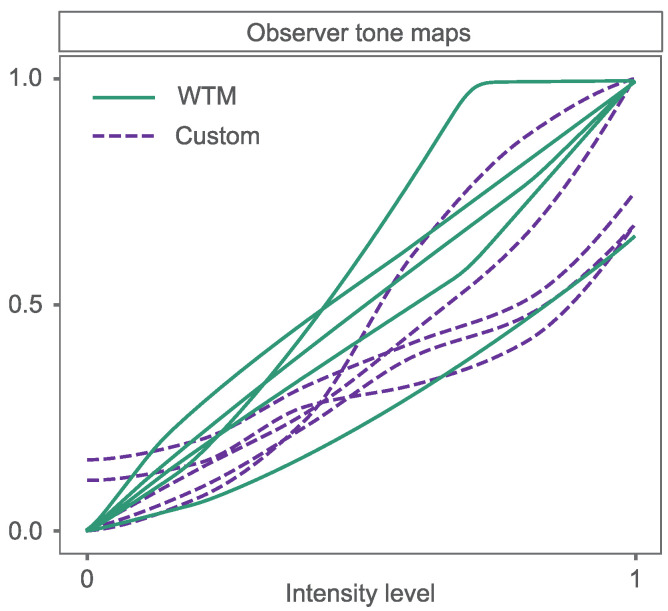
Example of observer tone-maps using WTM and the custom (control-point) enhancement. Custom tone-maps here could not be approximated by WTM i.e., they did not meet mean $\Delta E_{ab}^{*}$ (<5) threshold for approximation.

**Figure 13 sensors-23-03533-f013:**
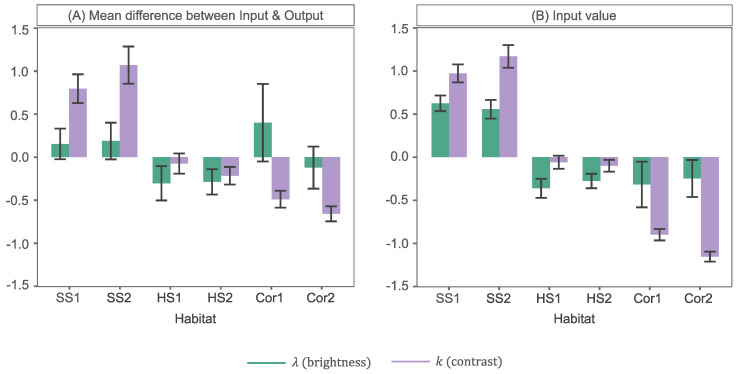
WTM parameter variation with image class (habitat): (**A**) mean difference in WTM parameters between input and output WD and (**B**) mean WD parameters across input images. All values are presented as standardised Z–Scores for fair comparison. The λ and *k* differences in (**A**) refer only to cases in which they were decreased and increased, respectively. Error bars represent 95% confidence intervals.

**Table 1 sensors-23-03533-t001:** Habitat classes encountered within the image dataset.

ID	Broad Habitat	Included Sub-Habitats
SS1	Soft Substrate (SS)	Heavily bioturbated SS, Single sea pen & Sea pen community
SS2	SS Sponge Community	-
HS1	Hard Substrate (HS)	Gravel area, Scattered Cobbles, Cobble and boulder area, Boulder area
HS2	HS Sponge Community	-
Cor1	Reef Framework	Coral rubble zone, Dead & Live *Desmophyllum pertusum* reef framework
Cor2	Soft Corals	Lone soft coral, Multiple soft coral colonies, HS soft coral community

**Table 2 sensors-23-03533-t002:** Mean image quality metrics for input images and enhanced outputs. Mean output metrics are also displayed according to enhancement type: WTM and custom control-point adjustment.

Image	UIQM	UCIQE	CCF
**Input**	0.68 (±0.02)	0.66 (±0.02)	7.84 (±0.2)
**Output**	**0.74 (±0.02)**	**0.68 (±0.02)**	**8.47 (±0.18)**
**Output${}_{\mathrm{WTM}}$**	0.72 (±0.02)	0.68 (±0.02)	8.29 (±0.19)
**Output${}_{\mathrm{custom}}$**	**0.85 (±0.04)**	**0.72 (±0.06)**	**9.18 (±0.47)**

1. 95% confidence intervals are shown in parentheses. 2. Best results are highlighted in bold.

**Table 3 sensors-23-03533-t003:** Intra-observer variability across image pairs.

Observer	1	2	3	4	5	6	7	8	9	10	Mean	Std
Agreement	1	1	1	1	0.61	0.94	0.94	1	0.94	0.83	0.93 (±0.08)	0.12
$MCV_{\lambda }$	0.11	-	0.06	0.07	0.05	0.08	0.08	0.08	0.09	0.03	0.07 (±0.01)	0.02
$MCV_{k}$	0.11	-	0.13	0.12	0.08	0.13	0.1	0.12	0.12	0.07	0.11 (±0.01)	0.02
$MCV_{c}$	-	0.19	-	-	0.28	-	-	-	0.14	0.06	0.17 (±0.09)	0.09

1. Agreement = proportion of matching decisions across image pairs. 2. *MCV_λ_*, *MCV_k_* = mean *CV* for *λ* & *k* in WTM enhanced pairs. 3. *MCV_c_* = mean *CV* for control-points in control-point enhanced pairs (*CV* here refers to mean *CV* across 6 control-points). 4. Hyphen denotes when enhancement tool was not selected. 5. 95% confidence intervals are shown in parentheses.

**Table 4 sensors-23-03533-t004:** Inter-observer variability across image pairs.

Agreement	${\mathrm{MCV}}_{\lambda }$	${\mathrm{MCV}}_{k}$	${\mathrm{MCV}}_{c}$
0.81 (± 0.04)	0.13	0.21	0.26

1. Agreement = proportion of matching decisions across image pairs. 2. *MCV_λ_*, *MCV_k_* = mean *CV* for *λ* & *k* in WTM enhanced pairs. 3. *MCV_c_* = mean *CV* for control-points in control-point enhanced pairs (*CV* here refers to mean *CV* across 6 control-points). 4. 95% confidence intervals are shown in parentheses.

**Table 5 sensors-23-03533-t005:** Summary statistics for mean $\Delta E_{ab}^{*}$ (across pixel pairs) between custom tone-mapped images and their WTM approximation.

Mean	% Images < 5	% Images < 1
2.98 (± 0.52)	91	11

1. 95% confidence intervals are shown in parentheses.

**Table 6 sensors-23-03533-t006:** Mean image quality metrics for custom tone-mapped images and their successful WTM approximations (mean $\Delta E_{ab}^{*}$< 5).

Image	UIQM	UCIQE	CCF
WTM approx.	**0.85 (±0.04)**	**0.72 (±0.06)**	8.94 (±0.43)
Custom	**0.85 (±0.04)**	0.71 (±0.06)	**9.14 (±0.48)**

1. 95% confidence intervals are shown in parentheses. 2. Best results are highlighted in bold.

## Data Availability

Commercial restrictions apply to the availability of these image data.

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
