# Peer review of "Weibull Tone Mapping (WTM) for the Enhancement of Underwater Imagery"

_sensors, 2023, doi:10.3390/s23073533_

Round 1

Reviewer 2 Report

In this paper, the authors designed a method to improve image quality using tone mapping operations (TMO) based on Weibull distributions. This kind of method is particularly important for improving the quality of images collected in underwater weak light environment. The authors have proved through experiments that this method can well help users get the image enhancement effect they want. After reading this paper, I have the following suggestions to improve the paper:

1. The authors have not summarized the innovation of this paper well. Perhaps the authors are explaining the innovation in the paragraph of Line 99-106? In this paragraph, the authors write: "The prior work on WTM [9,16] did not allow users themselves to enhance images according to the WTM theory. A key contribution of this paper is to allow such direct manipulation. ", but there are two problems here. First, the description of innovation is not clear enough. Usually we would explicitly list (1), (2), (3) and state what is the improvement over previous techniques. Secondly, the "prior work" mentioned by the authors here is just two conference papers published by the same authors, and both were published at unimportant conferences. In my opinion, they are not persuasive enough to be compared (forgive me for describing work published at unimportant conferences like this, but in general, the impact of a journal in some way reflects the importance of the work published in it). So perhaps the authors should do a better job of explaining what's innovative about the paper.

2. The authors' experiment lacked results that could be quantitatively compared. We all know that the effectiveness of an algorithm is generally evaluated from both qualitative and quantitative perspectives. The method studied by the authors belongs to the field of tone mapping operations (TMO), and there are many indicators that can be used to evaluate the results quantitatively, such as TMQI score and FSITM. Lack of such quantitative results will make the method unevaluable. Frankly, qualitative evaluation of results by "image analysts" or "observers" is not objective enough. If there are objective quantitative results that can prove the progressiveness of this method, it will be better.

3. The authors' experiments lack sufficient comparative methods. When we design an algorithm, we usually choose several comparison algorithms to test on the same data set to show that the method has indeed made a contribution to the field.The number of comparison algorithms may be two, may be eight, or other number, but it can't be without. For example, Panetta et al. [1] proposed a Parameter-Free Tone Mapping Operator called TMO-Net. In TABLE 1 of the paper, the authors compared their method with another 18 existing TMOs. Of course, 18 comparison methods seem too many. I think 3-4 methods may be necessary. As another example, Kim et al. [2] proposed a Tone Mapping Networks, and the author also showed five comparison algorithms in TABLE 1. I think such a comparison may be necessary.

[1] Panetta K, Kezebou L, Oludare V, et al. Tmo-net: A parameter-free tone mapping operator using generative adversarial network, and performance benchmarking on large scale hdr dataset[J].  IEEE Access, 2021, 9: 39500-39517.

[2] Kim H Y, Park S, Shin Y G, et al. Detail restoration and tone mapping networks for x-ray security inspection[J].  IEEE Access, 2020, 8: 197473-197483.

4. The author's Abstract could have been shorter. In fact, half of the content in Abstract is more suitable to be placed in the Introduction or Background section, so as to highlight the contribution of this paper.

5. Some possible typos, maybe the writer needs to check the spelling. I will give some examples:

Line 249: "the the corresponding CDF"? one of the 'the' may be redundant.

Line 287: "4. Experments", maybe it should be experiments.

6. In conclusion, the improvement and comparison of algorithm performance is necessary.

Round 2

Reviewer 2 Report

We believe that the author has revised the first opinion.